# Next-Generation Personalized Medicine: Implementation of Variability Patterns for Overcoming Drug Resistance in Chronic Diseases

**DOI:** 10.3390/jpm12081303

**Published:** 2022-08-10

**Authors:** Yaron Ilan

**Affiliations:** Department of Medicine, Hadassah Medical Center, Faculty of Medicine, Hebrew University, Jerusalem POB12000, Israel; ilan@hadassah.org.il

**Keywords:** personalized medicine, chronic disease, algorithms, chronic therapy, digital systems

## Abstract

Chronic diseases are a significant healthcare problem. Partial or complete non-responsiveness to chronic therapies is a significant obstacle to maintaining the long-term effect of drugs in these patients. A high degree of intra- and inter-patient variability defines pharmacodynamics, drug metabolism, and medication response. This variability is associated with partial or complete loss of drug effectiveness. Regular drug dosing schedules do not comply with physiological variability and contribute to resistance to chronic therapies. In this review, we describe a three-phase platform for overcoming drug resistance: introducing irregularity for improving drug response; establishing a deep learning, closed-loop algorithm for generating a personalized pattern of irregularity for overcoming drug resistance; and upscaling the algorithm by implementing quantified personal variability patterns along with other individualized genetic and proteomic-based ways. The closed-loop, dynamic, subject-tailored variability-based machinery can improve the efficacy of existing therapies in patients with chronic diseases.

## 1. Introduction

Treatment of chronic diseases is a significant health burden. Over a third of patients with various chronic diseases develop partial or complete resistance to chronic therapies, necessitating increased dosages, additional treatments, or replacing medications with newer, typically expensive drugs, which are not always associated with a better response [1].

Variability patterns are inherent to many biological systems [2]. We review the concept of using a closed-loop, dynamic, individualized variability-based machinery for improving the efficacy of existing drugs in patients with chronic diseases.

## 2. Treatment of Chronic Diseases Is a Significant Worldwide Health Burden

The World Health Organization (WHO) defines chronic disease as slow progression, long duration, and not transferable from person to person [3]. The Global Burden of Disease study reported a significant growth in the years lived with disability (YLD) over the last decade [4,5]. The prevalence of chronic condition multi-morbidity increases with age. About 40% of subjects over 44 suffer from a chronic disease. The percentages rise to 50% for 65–74-year olds and 70% for individuals over 85 years of age [5,6]. In 2014, 60% of Americans suffered from at least one chronic condition, and 42% had multiple chronic conditions [7]. Chronic diseases are the most significant cause of death globally, accounting for 7 of 10 deaths each year [8,9]. Cardiovascular diseases, hypertension, diabetes, stroke, depression, cancer, chronic respiratory diseases, epilepsy, fatty liver, obesity, mental health, and rheumatologic diseases denote some of the most common and costly chronic diseases [8,10,11]. Population aging leads to a surge in people suffering from chronic diseases [12]. The Center for Disease Control (CDC) estimates that 90% of the world’s USD 3.3 trillion in yearly health care expenditure is for individuals with chronic disease [13]. The direct cost of handling chronic disease was USD 1.1 trillion in the U.S. in 2016, equal to 5.8 percent of the gross domestic product (GDP) [14].

## 3. Drug Resistance Is Associated with Partial or Complete Loss of Effect and with Poor Prognosis in Patients with Chronic Diseases

Adaptation to prolonged exposure to drugs or development of any tolerance and tachyphylaxis prohibit the maximal and long-lasting effects of drugs [1]. Drug tolerance describes a subject’s reduced reaction to a drug after repeated use [15]. Resistance refers to the ability of a target subcellular, cellular, or whole organ to resist the effects of a drug that is usually effective against them [16]. The reduced response to medication following recurrent dosing reflects an adaptive body’s response to continued exposure [15]. While augmenting dosages may re-increase the response, it can accelerate the tolerance or increase side effects [17].

Several mechanisms underlie resistance, which is not identical for all drugs/subjects/diseases [18,19]. Adaptation and habituation to therapy can be at the levels of subcellular, cell organelles, and organs. The extent of adaptation depends on the disease, the drug, individual genetic and other factors, and the type of medication, dosage, and duration of treatment [20,21,22,23]. Circadian rhythms due to endocrinological or other mechanisms partly underlie drug resistance [24,25].

Pharmacokinetic or metabolic types of tolerance may occur due to the induction of drug-metabolizing enzymes. This type of tolerance happens due to a reduced amount of the substance reaching its target [26,27]. The therapeutic effect directed to an exact target at the molecular level depends on four variables: absorption, distribution, metabolism, and excretion (ADME). These parameters determine drug properties’ efficacy, duration of action, and side effects. The disturbance of one or more ADME elements can lead to drug resistance [28]. Polymorphic variants in several ADME genes are recognized for their role in toxicity [29,30]. The induction of enzymes is partially accountable for the tolerance; a recurrent drug administration reduces its outcome. For some medicines, metabolism with UDP-glucuronosyltransferases (UGTs) can transform drug substrates into polar metabolites, which are improved compounds for the various transporters compared with the original drug. UGT-linked resistance is related to enzyme overexpression. Studies showed that multidrug resistance is glucuronidation for medications used to manage hypercholesterolemia, hypertension, epilepsy, and psychiatric diseases [28]. The potential role of UGT1A1 *28/*28 in reducing the conversion of irinotecan to inactive metabolites requires testing and dosing adjustments based on the UGT1A1 genotypes before starting the chemotherapy schedule based on irinotecan [31].

Induction of CYP 450 enzymes by smoking (CYP1A2) and exposure to other drugs (CYP3A4) are additional examples of tolerance [32].

Pharmacodynamics or functional tolerance is due to “adaptation” of the drugs’ targets, such as losing receptor sensitivity. Repeated use reduces the cellular response to a substance. A common cause is increased compound concentrations continually binding to the receptor, desensitizing it via continual interaction [17]. A reduction in receptor density and mechanisms leading to action potential firing rates are additional causes. The majority of pharmacodynamic tolerance ensues following continued exposure to a medication. However, occurrences of rapid tolerance can occur [33]. Functional tolerance can end up with a complete loss of drug activity.

Behavioral tolerance happens following the administration of psychoactive drugs. In these circumstances, tolerance to a behavioral effect of a medication, for example, methamphetamine-associated increased motor activity, is a result of regular use. It happens due to a pharmacodynamic tolerance in the brain or via drug-independent learning [34]. Tachyphylaxis is an abrupt onset of tolerance to medications, which is not dose-dependent. It is a form of short-term tolerance following the dosing [35]. Opioid-induced hyperalgesia is another common form of tolerance [36].

Partial or complete drug resistance to chronic medications significantly impacts the outcome of chronic diseases and is associated with poor prognosis. Attempts to improve adherence to chronic therapy do not necessarily improve the disease outcome; they may increase non-responsiveness. Examples include the higher expenditures and hospitalization rates for patients with insulin resistance compared to non-insulin-resistant patients [37] and the increased mortality of patients with resistance to pain killers [38].

## 4. Drug Resistance in Common Chronic Diseases

Described below are typical examples of drug resistance that significantly impact both patients and society.

### 4.1. Loss of Effect for Antiepileptic Drugs

Epilepsy is diagnosed in more than 70 million people worldwide. Introducing new anti-seizure drugs did not assist one-third of patients who continue to suffer from seizures that are refractory to pharmacotherapy [39]. Nearly all first-, second-, and third-generation antiepileptic medications lose their efficacy, to some extent, during continued treatment [40]. The pharmacokinetics, target, intrinsic severity, gene variant, neural network, and transporter hypotheses may explain this phenomenon. Increasing doses, replacing the drugs, and brain stimulation and surgery are used in these patients with modest success [41,42,43].

### 4.2. Loss of Effect for Drugs That Work on the Heart and Blood Vessels

Nitrates effectively relieve angina pectoris symptoms in acute settings and prevent angina. However, their chronic use is associated with tolerance [44]. Tolerance to the effect of beta-blockers is well established following prolonged use. A dose-related loss of cardioselectivity of metoprolol has also been described [25,45].

### 4.3. Diuretic Resistance in Patients with Congestive Heart Failure Showing Diuretic Resistance

Heart failure (HF) is a growing epidemic associated with high morbidity, mortality, and healthcare expenditures. It affects 6 million people in the United States and represents a leading cause of cardiovascular mortality [46]. Volume overload symptoms are considered part of the morbidity, and reduced quality of life associated with HF. Loop diuretics are the mainstay in the treatment of volume overload. Diuretic resistance is a failure to reach congestion relief despite proper or escalating doses of diuretics [25,47]. Resistance to loop diuretics results from multiple variables associated with constant daily administration of equal or higher dosages [48]. Some of the measures taken in these patients are using higher dosages of diuretics, intravenous administration, or adding sequential nephron blockade by using combinations of two or more compounds from other classes of diuretics. These measures can further increase resistance [47].

### 4.4. Insulin Resistance in Subjects with Diabetes

Impaired response to exogenous insulin characterizes insulin resistance in type 2 diabetes mellitus. Patients who require more than 1 unit/kg/day are classified as having insulin resistance, and those requiring above 2 units/kg/day are classified as severe resistance. Alternatively, a daily insulin dose above 200 units is considered severe insulin resistance [49,50]. A growing number of subjects develop severe insulin resistance necessitating large doses of insulin [49]. Insulin resistance is due to an effect on the insulin receptor requiring an increase in insulin dose in patients with type 2 diabetes over time [51,52]. Treating subjects with severe resistance is a significant challenge as it is hard to control their glucose levels reasonably. Loss of the effect of glucagon-like peptide-1 (GLP-1) analogs, glucokinase activators, and dipeptidyl peptidase 4 (DPP4) inhibitors has been described in some patients [53,54].

### 4.5. Loss of Effect in Antidepressant and Anti-Psychotic Medications

Antidepressant resistance occurs in depressed subjects who lose their response to a formerly effective antidepressant therapy while receiving similar drugs and dosages [55,56]. Depression is classified as resistant when at least two attempts with antidepressants from different classes, proper doses, duration, and compliance do not achieve remission. A substantial proportion of subjects with depression ultimately develop treatment-resistant or refractory depression (TRD) [57,58]. The current therapy for TRD involves evaluating comorbid medical and psychiatric conditions and attempts to enhance antidepressant efficacy by augmentation, combination, or switching of anti-depressants [57,59]. However, a majority of TRD patients do not adequately respond. Anti-psychotic tolerance refers to a decrease in drug effect resulting from chronic use and the brain’s adaptive response to the foreign drug property [60]. “Drug learning and memory” notions were suggested to underlie the development of resistance, reflecting associative and non-associative processing that are affected by behavioral, environmental, and pharmacological variables. Drug-induced neuroplasticity, associated with functional alterations of the receptors and signaling pathways of prefrontal serotonin (5-HT)2A and striatal dopamine D2 receptors are possible mechanisms [60]. An “irregular” or “pulse therapy” approach might not be appropriate for some conditions, such as depression, where inconsistent drug use is a risk factor for disease relapse and other complications.

### 4.6. Loss of the Effect of Anti-Cancer Medications

Drug resistance in oncology is a multifactorial phenomenon [61]. Tumors show an acceptable response following the initial administration of chemotherapeutic drugs. However, they may become resistant after repeated treatments [62,63]. Cancer cells may resist chemotherapy due to spontaneous mutations in any growing cell subset, whether exposed to treatment or not [17]. When a drug destroys healthy cells, a higher percentage of the survivors may become resistant. Pharmacogenomic, pharmacokinetic, pharmacodynamic factors, and drug selection and dosing determine resistance. Exosomal miRNAs also contribute to the development of drug resistance [62]. Epidermal growth factor receptor (EGFR) inhibitors, panitumumab, and cetuximab used to treat metastatic colorectal cancer expressing wild-type *KRAS* show a beneficial effect in only 10–20% of patients [64,65]. Drug resistance limits the effectiveness of therapies for lung cancer and is associated with tumor heterogeneity [66]. The resistance acquired after partial administration of chemotherapy is associated with developing more aggressive clones. Mutations may be spontaneous but are induced mainly by exposure to drugs [61]. Intratumoral heterogeneity contributes to the inconsistency of responses to chemotherapy, which is not captured by most current approaches based on cancer cell biomarkers. Some contributing factors to the development of drug resistance are as follows: development of alternative pathways for growth activation; interpatient variability of genetic and epigenetic factors; altered differentiation pathways; alterations in drug targets; alterations of the local physiology of tumor, such as blood supply; behavior of neighboring cells; the anti-tumor immune response; drug transport; intracellular distribution; and apoptosis inhibition and increased enzymatic activity [67,68].

### 4.7. Loss of Effectivity in Treating Neurological Disorders

Disease-modifying therapies for multiple sclerosis, such as immunomodulatory drugs, have high variability in efficacy. Individual response to current medications significantly varies across patients; moreover, 30–80% discontinue therapy [69,70].

### 4.8. Loss of Effect for Painkillers

Drug tolerance to chronic analgesics necessitates increasing dosages [14]. Following tobacco, alcohol, and marijuana, the most commonly abused drugs are methylphenidate (Ritalin), Diazepam (Valium^®^), and oxycodone (OxyContin^®^). Opiates are the primary approach for chronic pain therapy in cancer patients suffering from moderate to severe pain and in patients with non-cancer chronic pain. Patients with chronic pain often develop tolerance to opioids over time with aggravated pain [71]. Prolonged administration of opiates is associated with developing antinociceptive tolerance. Higher doses are mandatory overtime for reaching the same degree of analgesia [72]. Sustained exposure to morphine results in paradoxical pain, which occurs in regions not affected by the original pain resulting in dose escalation, termed ‘analgesic tolerance’ [72,73]. Narcotic medications downregulate the Mu receptors in the brain. With fewer receptors, it takes more narcotic-like molecules for subjects to obtain the same feeling. This down-regulation leads to tolerance and a need for increased dosages over time to achieve pain relief [74,75,76].

### 4.9. Loss of the Effect of Immunomodulatory and Anti-Inflammatory Drugs

Partial or complete loss of anti-TNF drugs’ impact occurs in patients with inflammatory bowel diseases (IBD), rheumatoid arthritis, and psoriasis [77,78]. Loss of response (LOR) to anti-TNF therapy is common in IBD patients [79]. The incidence of LOR among adult IBD patients is 36% [80,81,82]. About 50% of subjects with Crohn’s disease or ulcerative colitis develop LOR to infliximab following an early response to the drug [82]. There is no difference in time to LOR between subjects treated using regimens comprising several drugs or different anti-TNF agents [83]. Switching between anti-TNF agents, dose intensification, and adding an immunomodulator to suppress immunogenicity may overcome LOR with moderate success [84,85].

## 5. Averages-Based Treatment Regimens Are Associated with Non-Responsiveness, and Current Measures of Personalized Medicine Are Insufficient to Overcome It

Treatment of chronic diseases commonly follows a pre-determined regimen. It is carried out based on protocols within the therapeutic and efficacy windows. Once a treatment regimen is prescribed/configured, it stays identical until complete or non-responsiveness occurs. Most drugs and therapies developed for the “average patient” are insufficient for most subjects. However, patients respond differently to similar treatments. Using averages to determine drugs’ effects leaves high proportions of patients as partial or complete non-responders [86,87,88]. Only a relatively small number of subjects respond to the medication, exposing others to the risk of side effects through ineffective therapies [89]. Using averages in medicine is insufficient for developing personalized approaches. However, there are no valid methods for most diseases to determine which treatment is best for an individual patient [1,90].

Differences in multiple parameters drive inter-individual variation in drug response. Gene interactions are studied using gene regulatory networks, RNA velocity, and single-cell sequencing of thousands of cells. Single-cell data combined with big data can reconstruct personalized, cell-type, and context-specific gene regulatory networks [89,91]. However, environmental exposures, proportions of cell types involved, patients’ genetic background, and variabilities in proteomics and metabolomics are nearly impossible to control simultaneously.

Therefore, attempts to use “personalized” measures, such as genomics, proteomics, metabolomics, and others, are only partially successful in overcoming the challenge of improving response to chronic therapies.

## 6. Variability Is Inherent to Biological Systems: Loss/Change in Variability Patterns Leads to Poor Prognosis

Intra- inter-patient variabilities are inherent to biological systems, with their dynamics affected by both intrinsic and extrinsic sources [92,93]. These systems manifest a high rate of uncertainties regarding the specific sources of variabilities evolving from multiple genetic, biochemical, and metabolic variables [94,95,96,97].

Variability exists to form the levels of genes and molecules to that of whole organs [2,94,95,96,98,99,100,101]. The stochastic behavior resulted from collisions among molecules within the entire cellular compartment and characterized all living systems [102]. Intrinsic and extrinsic stochasticity leads to single-cell variability in gene expression—the intrinsic stochasticity results from randomness in gene expression processes and mRNA and protein synthesis pathways. Extrinsic alterations mirror the status of systems and their communications with the intracellular and extracellular environments [102].

Variability is the hallmark of microtubules’ function. The microtubules constitute the cellular cytoskeleton, and their dynamic instability is a feature of biological variability that characterizes their function. Their dynamic behavior constitutes the basis for multiple biological processes contributing to cellular plasticity and cell signaling [100,101].

Cell death processes manifest variability. Not all cells expire at the same treatment dose or at the same time of a chemotherapeutic drug. This cell-to-cell variability results from differences in apoptosis signaling networks and intracellular and extracellular parameters [103]. Intra- and inter-cell variabilities occur to express cell epitopes, cell singling pathways, and cytokine secretion, suggesting a marked heterogeneity in cells from the same person [91].

Variability is inherent to the function of whole organs. Patterns of variability are associated with normal physiology and health. Examples of variability inherent to healthy organs’ functions include variations in heart rate [104], breathing [105,106], gate, and blood glucose levels [107]. Changes in normal physiologic variability loss lead to disease states and bad outcomes [107,108,109,110].

## 7. Intra- and Inter-Patient Variabilities for Pharmacodynamics, Drug Metabolism, and Response

A high inter- and intra-patient variability characterizes drug responsiveness, metabolism, and pharmacodynamics. This variability is associated with partial or even complete loss of drug effectiveness [111,112,113,114]. The cell is crowded with uneven distribution of macromolecules that interrelate with a drug in multiple specific and non-specific ways. This phenomenon results in a high heterogeneity in drug response between cells [115]. A study showed marked daily variability in antiepileptic levels in subjects stabilized with the same drug over time [116]. High intra-patient variability to tacrolimus, an immunosuppressive drug, was associated with graft rejection [114].

Interindividual variability in drug efficacy and toxicity is a significant challenge in designing personalized therapeutic regimens. Variations in drug pharmacokinetics (PKs) and pharmacodynamics (PDs) are partially a result of the polymorphic variants in genes encoding drug metabolizing enzymes and transporters and genes encoding drug receptors. Pharmacogenomics (PGx) assists in selecting biomarkers of the pharmacology variables of genome-drug interactions enabling personalizing therapies [117].

Regular drug dosing regimens are incompatible with physiological variability and may contribute to resistance to the effect of medications. A “drug holiday” can sometimes overcome tolerance [118]. The successful design of therapies necessitates looking into the intrinsic variabilities in the responses to medications [1,119].

## 8. Applying Variability to Biological Systems

The chaos theory describes deterministic systems with predictable behavior; however, these systems may become random [120,121]. The theory focuses on activities of non-linear dynamic systems that are sensitive to the primary settings [122,123,124]. In chaotic systems, minor differences in the opening conditions may lead to different outcomes. Whether these systems are deterministic, their long-term behavior prediction is difficult [125].

The interplay between disorder and order in a chaotic system is a significant task in biology [94,95,96]. Chaos theory enables quantifying the degree of order in biological systems. Mathematical formulations based on chaos theory are somewhat appropriate to biology [126]. Several concepts of this theory, including fractal dimension, entropy, and algorithms for obtaining quantitative characteristics of the degree of the order, have been applied [127].

Based on the complexity and chaos theories, systems biology applies models to forecast the associations between genes, proteins, and different variables in the external milieus of these systems [128]. Network medicine further adapts systems biology to clinical sciences [128]. Combining high-throughput data collection within molecular or higher-order systems modeling can improve the results [129]. Data-driven techniques can determine chaotic and random systems and assess their dynamics [94,95,96,130,131]. Network models can establish scale-free features, reproducing the networks’ structures for the interactions between proteins [132]. Studies showed this concept applies to several physiological processes, including neuronal activity, breathing, heart rate, and electroencephalograms [127].

Recent studies described the chaotic modeling approaches from the molecular to whole organs, including cardiac rhythms, brain dynamics systems [133], eye tracking disorder in schizophrenics [134], and warning signs of fetal hypoxia [135]. Genome chaos is the process of complex, rapid genome re-organization, leading to chaotic genomes, followed by establishing stable genomes [91,136]. Mathematical models predict calcium oscillations in vascular smooth muscle [137]. An algorithm of the bone-density stress adaptation model involves a chaos mechanism [138]. Chaos approaches helped uncover a peptide or RNA state of integrated protometabolism. Chaos theory revealed the mechanism that drives genetic heterogeneity observed in tumors [139].

## 9. A Three-Phase Roadmap for Developing a Platform for Overcoming Drug Resistance in Patients with Chronic Diseases

Recent studies described the development of a platform for overcoming drug resistance based on the following three phases [1,140,141]: introducing irregularity in therapeutic regimens for improving drug responsiveness; establishing a closed-loop algorithm for generating individualized patterns of irregularity to overcome drug resistance; upscaling the algorithm by implementing quantified personal variability patterns along with additional personalized signatures based on genes, proteins, and other disease or host relevant variables [1,25,43,68,76,85,98,99,100,101,142,143,144,145,146,147,148,149,150,151].

Figure 1 illustrates a three-step approach for introducing a system for overcoming resistance to chronic drugs.


Step A: Using irregularity to overcome drug resistance: Implementing treatment regimens based on aperiodic regimens of taking the drugs at irregular intervals and strengths.Step B: Establish a closed-loop algorithm for generating individualized patterns of irregularity for overcoming drug resistance: The closed-loop algorithm provides a method for overcoming the loss of response to drugs by setting up an irregularity within a specific range that is determined in a subject-tailored way. The algorithm reaches a physiological target. The algorithm receives inputs from the user and other users to update the treatment regimen.Step C: Upscaling the algorithm by implementing quantified personal variability patterns and additional personalized signatures: Cellular and whole organ patterns of variability are quantified and implemented into the algorithm. A dataset of variability patterns at the cellular levels using single cell-based techniques is generated from cells harvested from patients before and after chronic disease therapy, including single-cell RNA sequencing, proteomics, metabolomics, and epitope expression. Methods for quantifying these inherent variability patterns and combining single cells with whole organ variability patterns generate an individualized factor implemented into the treatment algorithm. The quantified variability patterns are implemented into individualized-based treatment regimens. The algorithm continuously alters the irregular regimen based on the patient’s closed-loop feedback on the therapy’s effect.


### 9.1. Using Irregularity to Overcome Drug Resistance

Regular administration of a constant daily dose, or a continuous increase in drug dose, is more likely to be associated with resistance to medications when compared with irregular administration of the same or altering dose [152]. Using treatment regimens based on nonperiodic routines of drug administration, and using variability in the intervals and strengths can reduce the likelihood of resistance and improve drug effectiveness [152].

A recent study described a pre-clinical model documenting variability in response to immunomodulatory drugs. Using the immune-mediated hepatitis model of Concanavalin A (ConA) and treatment with two immunomodulators (e.g., anti-CD3 or glucosylceramide (GC) were studied [142]. In multiple consecutive studies, the study showed an individualized response pattern of reaction to an injection of ConA and oral administration of immunomodulatory agents. As measured by liver enzymes, improvement of the liver injury showed marked intra-group and inter-experiment variabilities. Similarly, the data showed marked variability in the response for serum cytokine levels and lymphocyte subsets. Using irregularity in administering steroids and anti-LPS antibodies was superior in alleviating immune-mediated damage in this model compared to regular drug administration [142].

Clinical trials are ongoing to determine the use of irregularity for overcoming drug resistance in patients who have lost their response to drugs [25,43,68,76,85,143,144,145,146,147,148,149,150,151,153,154]. Patients are treated according to a regimen that introduces irregularity in the dose and intervals of drug administration, maintaining them within the drug’s therapeutic window and a physician’s pre-determined range [141].

### 9.2. Establishing a Closed-Loop Algorithm for Generating Individualized Patterns of Irregularity for Overcoming Drug Resistance

One research challenge is implementing subject-specific, disease-tailored, and drug-tailored features within closed-loop deep machine learning algorithms. The closed-loop algorithm provides a platform for reducing the loss of drug efficacy by setting up an irregularity within a specific range determined in an individualized form [141]. The algorithm aims to reach a physiological goal or target for each patient and disease. The patients or their care providers update the machine with inputs indicative of progress towards the desired goal. The learning machine provides updated dose and administration time parameters based on the data learned. It receives inputs from the user and other users to update the algorithm to enable redirecting or further define the treatment regimen [1,140,141].

### 9.3. Upscaling the Algorithm by Implementing Quantified Personal Variability Patterns along with Additional Personalized Signatures

It is essential to establish and quantify cellular and whole organ patterns of variability and then implement them into the algorithm for improving accuracy [25,43,68,76,85,97,100,101,141,142,143,144,145,146,147,148,149,150,151,153,154,155,156,157,158]. Important information is also obtained by evaluating single-cell variabilities in gene expression, proteomics, metabolomics, and epitope expression performed on cells harvested from patients before and after therapy for chronic disease. A dataset of variability patterns at the cellular levels using single cell-based techniques is generated from cells harvested from patients before and after chronic disease therapy, including single-cell RNA sequencing, proteomics, metabolomics, and epitope expression [159,160,161,162].

Quantifying variability signatures at the molecular level, such as the dynamic instability of microtubules, cytokine secretion, and others. These platforms can be used to implement novel therapies, improve response to chronic cytokine microtubules, overcome drug resistance, exert gut-based systemic immune responses, and generate patient-tailored dynamic therapeutic regimens [68,100,141].

Similarly, assessment of whole organ variability patterns also provides vital data. For example, patients with congestive heart failure are followed for their heart rate variabilities over a 24 h measurement period [25,163,164]. Patterns of variability changes are defined and correlated with patient clinical status and by comparing patients’ current pattern of heart rate variability to heart recordings before deterioration. Similarly, diabetes patients are followed for glucose level variabilities, and their patterns of variabilities were defined. An analysis of the data received from continuous glucose monitor (CGM) sensors in patients with type 2 diabetes showed that the variations in CGM data analyzed every 5 min are not simply “uncorrelated noise”. Quantification of the complexity of the CGM time series temporal structure using multiscale entropy analysis showed that the fluctuations in serum glucose levels from control subjects are more complex when compared with the data from patients with type 2 diabetes [107]. Methods for quantifying these inherent variability patterns, and combining single cells with whole organ variability patterns, must be established for generating an individualized factor to be implemented into the treatment algorithm.

The quantified inherent variability patterns, along with other personalized ways based on genomics, proteomics, microbiome-based, and different signatures, are implemented into individualized-based treatment regimens for overcoming drug resistance. The algorithm continuously alters the irregular regimen based on the patient’s closed-loop feedback on the therapy’s effect. The generated insightful database evolves and supports the dynamic individualized-irregular treatment regimens [1,140,141].

Study limitations: This paper does not present the results of using the described platform in patients with chronic disease. Ongoing studies will shed light on its use for developing personalized therapeutic regimens for patients with chronic diseases.

In summary, overcoming the partial or complete loss of responsiveness to chronic drug use is a significant health problem linked with high morbidity and mortality. The described closed-loop, deep machine learning algorithms generate an individualized, dynamic irregularity, which implements quantified variability and other personalized patterns into therapeutic regimens. This method may answer the unmet need for next-generation personalized medicine enabling a sustainable long-term effect for therapies targeting chronic diseases.

## Figures and Tables

**Figure 1 jpm-12-01303-f001:**
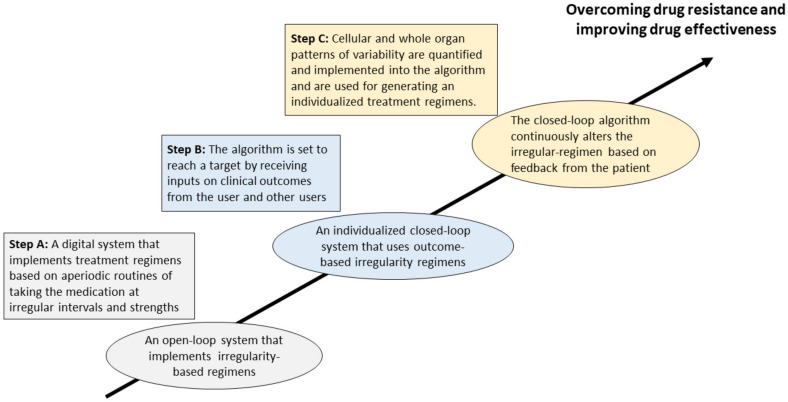
A three-step approach for introducing a system for overcoming drug resistance.

## Data Availability

All data are available in public resources.

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
