# Peer review of "Next-Generation Personalized Medicine: Implementation of Variability Patterns for Overcoming Drug Resistance in Chronic Diseases"

_jpm, 2022, doi:10.3390/jpm12081303_

Round 1
Reviewer 1 Report
Thank you for the invitation to review this manuscript. The author has tried to propose a three-phase platform to overcome resistance among chronic disease patients. To my knowledge, several studies have suggested various models addressing the same issue and how this review differs from other studies. I have no concern with the proposed suggestions in this study, but still, there is a need to affirm the rationale of this manuscript. Similar platforms have been well discussed to address antimicrobial resistance. There is a need to extend the introduction section to answer these questions. Please add the limitation section in the manuscript, as this is a narrative review and lacks the power to contribute to the literature. Please provide the conclusions section separately, with future research and practice directions.
Author Response
The author thanks the reviewer for the comments.
- In light of the comment, the Introduction section was revised. The section on differentiating the proposed platform from other systems was expanded and the relevant references were included.
- A study limitation section was included.
- A conclusion section was included.
Reviewer 2 Report
Dr. Ilan in this review described a three phase platform for overcoming drug resistance in order to avoid the lack of therapeutic effect. The manuscript has a limited novelty and the interest for a broader audience is limited. Major revisions are suggested for new consideration.
I evidenced the lack of a paragraph describing the methodological challenges and the emerging technological tools in PGx biomarker discovery and validation as for example well described in https://doi.org/10.1111/cts.12869.
Moreover, in the paragraph 2.2 no correlation was evidenced on the role of UGT1A1 *28/*28 reducing the conversion of irinotecan to inactive metabolites and which should need testing and dosing adjustments based upon the UGT1A1 genotypes before starting with chemotherapy schedule based on irinotecan.
I suggest also an inplementation of examples on polymorphic variants in several ADME genes recognized for their role in toxicity/ efficacy in the treatment of all diseases discussed in this review.
Also the description of the three phase platform probably should benefit of some applicative example of cases of pharmacogenetics problems solved using these approaches as well as a connection to available software for data analysis.
Author Response
The author thanks the reviewer for the valuable comments.
- The paper on genotyping is important. The relevant paragraph was updated to describe the use of PGx biomarkers. We have also included it as part of the description of the algorithm.
- We accept the comment on the UGT. The relevant paragraph was revised accordingly and the references were updated.
- We accept the remark on ADME. The paragraph was revised to include the relevant references on the role of the variants in drug toxicity.
- The section on the three steps algorithm was revised to include the pharmacogenetics data as suggested.
Reviewer 3 Report
In this manuscript , the author reviewed the intra- and inter-patient variability on the loss of drug effectiveness in management of chronic diseases, and proposed a three-phase platform for overcoming drug resistance. Overall the contents are excellent and the informative. Especially the different mechanisms of drug resistance among different drugs/diseases can be very helpful for researchers in this field. While the proposed platform is promising, the challenge is: how to implement those technologies in clinical practice? I feel we still have a long way to go there.
Author Response
We thank the reviewer for his comments.
In light of the remark the relevant paragraph was updated to tone down some of the conclusions made, and we also included a study limitation paragraph.
Round 2
Reviewer 2 Report
The authors satisfied all my suggestions.
Author Response
We thank the reviewer for his comments.